# Invasive Phenoprofiling of Acute-Myocardial-Infarction-Related Cardiogenic Shock

**DOI:** 10.3390/jcm12185818

**Published:** 2023-09-07

**Authors:** Jorge A. Ortega-Hernández, Héctor González-Pacheco, Jardiel Argüello-Bolaños, José Omar Arenas-Díaz, Roberto Pérez-López, Mario Ramón García-Arias, Rodrigo Gopar-Nieto, Daniel Sierra-Lara-Martínez, Diego Araiza-Garaygordobil, Daniel Manzur-Sandoval, Luis Alejandro Soliz-Uriona, Gloria Monserrath Astudillo-Alvarez, Jaime Hernández-Montfort, Alexandra Arias-Mendoza

**Affiliations:** 1Instituto Nacional de Cardiología Ignacio Chávez, Coronary Care Unit, Juan Badiano 1, Sección XVI, Tlalpan, Ciudad De Mexico 14080, Mexico; jardiel.arguello@gmail.com (J.A.-B.); j.omar.arenas@gmail.com (J.O.A.-D.); rpl9017@hotmail.com (R.P.-L.); mario.aris7@gmail.com (M.R.G.-A.); rodrigogopar@gmail.com (R.G.-N.); danielsierralaram@gmail.com (D.S.-L.-M.); dargaray@gmail.com (D.A.-G.); drdanielmanzur@gmail.com (D.M.-S.); luissolizu@gmail.com (L.A.S.-U.); monse_toons@hotmail.com (G.M.A.-A.); aariasm@yahoo.com (A.A.-M.); 2Advanced Heart Failure and Recovery Program for Central Texas Baylor Scott & White Health, 302 University Blvd, Round Rock, TX 78665, USA

**Keywords:** acute myocardial infarction, cardiogenic shock, pulmonary artery catheter, hemodynamic profiles, CS phenotypes, congestion profiling

## Abstract

Background: Studies had previously identified three cardiogenic shock (CS) phenotypes (cardiac-only, cardiorenal, and cardiometabolic). Therefore, we aimed to understand better the hemodynamic profiles of these phenotypes in acute myocardial infarction-CS (AMI-CS) using pulmonary artery catheter (PAC) data to better understand the AMI-CS heterogeneity. Methods: We analyzed the PAC data of 309 patients with AMI-CS. The patients were classified by SCAI shock stage, congestion profile, and phenotype. In addition, 24 h hemodynamic PAC data were obtained. Results: We identified three AMI-CS phenotypes: cardiac-only (43.7%), cardiorenal (32.0%), and cardiometabolic (24.3%). The cardiometabolic phenotype had the highest mortality rate (70.7%), followed by the cardiorenal (52.5%) and cardiac-only (33.3%) phenotypes, with significant differences (*p* < 0.001). Right atrial pressure (*p* = 0.001) and pulmonary capillary wedge pressure (*p* = 0.01) were higher in the cardiometabolic and cardiorenal phenotypes. Cardiac output, index, power, power index, and cardiac power index normalized by right atrial pressure and left-ventricular stroke work index were lower in the cardiorenal and cardiometabolic than in the cardiac-only phenotypes. We found a hazard ratio (HR) of 2.1 for the cardiorenal and 3.3 for cardiometabolic versus the cardiac-only phenotypes (*p* < 0.001). Also, multi-organ failure, acute kidney injury, and ventricular tachycardia/fibrillation had a significant HR. Multivariate analysis revealed that CS phenotypes retained significance (*p* < 0.001) when adjusted for the Society for Cardiovascular Angiography & Interventions score (*p* = 0.011) and ∆congestion (*p* = 0.028). These scores independently predicted mortality. Conclusions: Accurate patient prognosis and treatment strategies are crucial, and phenotyping in AMI-CS can aid in this effort. PAC profiling can provide valuable prognostic information and help design new trials involving AMI-CS.

## 1. Introduction

Acute myocardial infarction complicated by cardiogenic shock (AMI-CS) has a looming prognosis. Recently, the Society for Cardiovascular Angiography & Interventions (SCAI) classification offered new terms that clinicians and researchers can use to communicate the severity of the CS [1].

While the severity of CS can be assessed using the SCAI system, there is still a need for more comprehensive phenotyping of CS to understand its underlying mechanisms and improve patient outcomes. Recently, a phenotypification of CS into three clusters was provided by Zweck et al. [2,3]. Although SCAI 2022 provides an abstract form of measuring severity, the presence of organ involvement in the phenotyping process must be included. Also, dynamic congestion is paramount in AMI-CS outcomes [4]. Furthermore, hemodynamic trajectories must be studied because much of the information used to classify patients is single-point data, rather than with repeated measures [2,5,6].

We propose an analysis of the hemodynamic “signature” of AMI-CS using repeated measures data from pulmonary artery catheter (PAC) studies that could provide a more precise and individualized prognostication [7]. Furthermore, by identifying distinct hemodynamic patterns and their associations with CS phenotypes, we can improve our current understanding of AMI-CS heterogeneity and its potential complications.

## 2. Materials and Methods

We analyzed retrospective PAC data from 309 AMI-CS patients from January 2006 to July 2021 at the National Institute of Cardiology in Mexico City. CS was defined as a systolic pressure of ≤90 mmHg, the need for vasoactive or mechanical support (MCS), lactate ≥2 mmol/L, and/or a cardiac index of ≤2.2 L/min/m^2^ [8]. Our institution’s Research and Ethics Committee approved the study protocol, and patient consent was not required. All procedures were conducted in accordance with the Declaration of Helsinki and local regulations.

### 2.1. Classifications and Definitions

We classified the patients according to the 2022 SCAI stage, with the worst stage at 24 h after a PAC was installed [1]. We created a congestion profile for all patients using the standardized cut-off values [6] of pulmonary capillary wedge pressure (PCWP, ≥18 mmHg) and right atrial pressure (RAP, ≥12 mmHg), and we used the changes in these congestion profiles to categorize the patients into 3 groups: decongestive, neutral, or congestive within the first 24 h [1,2,4] (See Full Description in the Appendix A).

### 2.2. Cardiometabolic Phenotype

Three phenotypes were created based on parameters described by Zweck et al. [2] at the initial CS presentation: cardiac-only (no organ involvement), cardiorenal (estimated glomerular filtration rate (eGFR) < 60 mL/min/1.73 m^2^), and cardiometabolic (renal + hepatic (alanine transaminase (ALT)) 150 U/L, >3 upper normal limit) (For a full description, see the Appendix A).

### 2.3. Complications

Complications included ventricular tachycardia/fibrillation; acute kidney injury (AKI), defined by the KDIGO AKI guidelines (“KDIGO Clinical Practice Guideline for Acute Kidney Injury,” 2012) [9]; and multi-organ failure (MOF), defined by the Multiple Organ Dysfunction score (MODS) [10] for in-hospital 30-day follow-up (For a full description, see the Appendix A).

## 3. Statistical Analysis

The demographic data for qualitative variables were presented as frequencies and percentages, and the Chi-squared or exact Fisher tests were used as appropriate to assess the differences. For continuous variables, median and interquartile ranges and comparisons were performed using the Kruskal–Wallis or Mann–Whitney *U* tests for group comparisons. Bonferroni correction was used when the comparison involved multiple groups.

We used the ANOVA for repeated measures to evaluate the changes over time, and the Mauchly test was performed for the sphericity test. Finally, the Greenhouse–Geisser test was used to correct the degrees of freedom and to compare each group’s hemodynamic variables.

Kaplan–Meier curves were constructed, a 30-day restricted mean survival time (RMST) was obtained, and differences were assessed using the log-rank test. The phenotypes were compared to the SCAI score and congestion changes to evaluate the value of using the triple-scoring system to predict mortality. Furthermore, the phenotypes and outcomes were compared against in-hospital mortality in multivariate analyses against age, sex, DM, HTN, type of myocardial infarction (MI), primary reperfusion, and SCAI score. Finally, hazard ratios and their 95% confidence intervals were reported.

All statistical tests were 2-tailed, and significance was assumed if a *p*-value < 0.05 was obtained. Statistical analyses were performed using IBM SPSS Statistics, MedCalc for Windows (v19.4; MedCalc Software, Ostend, Belgium), and SAS on Demand for Academics (SAS Institute, Cary, NC, USA).

## 4. Results

The phenotypes were cardiac (43.7%), cardiorenal (32%), and cardiometabolic (24.3%). The cardiac group was the youngest compared with the cardiorenal and cardiometabolic groups (*p* < 0.001). The prevalence of hypertension was higher in the cardiorenal and cardiometabolic compared with the cardiac group (59.6% and 58.7% vs. 42.2%; *p* = 0.026 and 0.012, respectively). The incidence of previous chronic kidney disease was also higher in the cardiorenal and cardiometabolic compared with the cardiac group. The proportion of patients with diabetes mellitus was also higher but not significantly different in the cardiorenal group (*p* = 0.115).

Previous MI, percutaneous coronary intervention (PCI), and coronary artery bypass graft (CABG) were not significantly different among the three groups. There was a significant difference in the type of AMI (*p* = 0.022), with a higher percentage of STEMI in the cardiometabolic group (93.3%) compared with the other groups. The out-hospital cardiac arrest was not different (*p* = 0.92). The Killip–Kimball classification showed significant differences (*p* < 0.001), with a higher proportion of class IV in the cardiometabolic group.

The results show that the cardiometabolic group had the lowest left-ventricular ejection fraction (LVEF) of 30% compared with 35% in the cardiac and 32% in the cardiorenal group; however, this was not statistically significant (*p* = 0.073).

White blood cell counts were higher, and platelets were lower, in the cardiometabolic group compared to the cardiorenal and cardiac groups. Glucose levels were significantly higher in the cardiometabolic compared with the cardiac (*p* = 0.02) and the cardiorenal groups (*p* = 0.034). BUN and creatinine levels were significantly higher in the cardiorenal group (*p* < 0.001). The eGFR was considerably lower in the cardiorenal and cardiometabolic groups (*p* < 0.001).

Values of aspartate transaminase (AST), ALT, lactate dehydrogenase (LDH), maximum creatinine, maximum AST, maximum ALT, bilirubin, and lactate significantly differed in the groups, with the highest values found in the cardiometabolic group. The cardiometabolic group had lower base excess and pH values (*p* < 0.001).

In primary reperfusion cases, the rate of thrombolysis was 20.1% (27) in the cardiac, 17.2% (17) in the cardiorenal, and 6.7% (5) in the cardiometabolic groups. The rate of primary percutaneous coronary intervention (pPCI) was 34.3% (46) in the cardiac, 27.3% (27) in the cardiorenal, and 21.3% (16) in the cardiometabolic group. In comparison, the rate of non-primary reperfused cases was 45.5% (61) in the cardiac, 55.6% (55) in the cardiorenal, and 72% (54) in the cardiometabolic groups (*p* = 0.005). (For specific coronary artery distribution in different cardiogenic shock profiles, refer to Appendix A).

Likewise, the rate of mechanical ventilation was 60.7% in the cardiac, 72.7% in the cardiorenal, and 80% in the cardiometabolic groups, and the difference was statistically significant (*p* = 0.01). Finally, the rate of hemodialysis was 4.4% (6) in the cardiac, 17.2% (17) in the cardiorenal, and 28% (21) in the cardiometabolic groups (*p* < 0.001).

Norepinephrine was used in 74.1% of the cardiac, 82.8% of the cardiorenal, and 88% of the cardiometabolic groups (*p* = 0.039). Vasopressin was used in 45.9% of the cardiac, 56.6% of the cardiorenal, and 73.3% of the cardiometabolic groups (*p* = 0.001). Levosimendan was used in 25.9%, 27.3%, and 48% of the cardiac, cardiorenal, and cardiometabolic groups, respectively (*p* = 0.002). Dobutamine use was lower in 76.3% of the cardiac group compared with 83.8% and 97.3% of the cardiorenal and cardiometabolic groups, respectively. Considering the number of vasoactive drugs, a higher proportion of the cardiometabolic group had four drugs compared with the cardiac and cardiorenal groups (*p* < 0.001).

No differences were seen regarding mechanical circulatory support (MCS), with the majority being intra-aortic balloon pumps, with only two patients with a cardiorenal phenotype receiving extracorporeal membrane oxygenation and only two patients receiving Impella-CP (one each in the cardiorenal and cardiometabolic groups, respectively).

The incidence of AKI was higher in the cardiorenal (82.8%) and cardiometabolic (92%) groups compared with the cardiac group (43%) (*p* < 0.001). Furthermore, the severity of AKI was higher in the cardiometabolic and cardiorenal compared with the cardiac group (*p* < 0.001).

Significant differences existed in the number of organ failures and the presence of MOF. A higher percentage of patients with cardiometabolic comorbidities had MOF than in the other groups (80% vs. 62.6% in the cardiorenal and 41.5% in the cardiac group; *p* < 0.001). In addition, a higher percentage of patients with the cardiometabolic phenotype had ≥4 more organ failures than in the other groups (33.3% vs. 18.2% in the cardiorenal and 6.7% in the cardiac groups; *p* < 0.001).

The cardiometabolic group had higher MODS scores, indicating higher organ dysfunction levels than in the other groups (*p* < 0.001). The same pattern was seen with the SCAI score, with patients in the cardiometabolic and cardiorenal groups having higher scores (*p* < 0.001).

Finally, the highest mortality rate was seen in the cardiometabolic (70.7%), followed by the cardiorenal (52.5%) and cardiac groups (33.3%), with significant differences (*p* < 0.001; see Table 1 and full pairwise comparison in Appendix A).

### 4.1. Hemodynamic Variables and CS phenotypes 

Heart rate did not show any significant differences in the three groups as a whole and at any time point (*F* = 0.44; *p* = 0.644), nor was any interaction seen in the within-subjects’ effects.

However, SBP showed significant differences between subjects, where the effects were lower in the cardiometabolic and higher in the cardiac groups at all time points (*F* = 3.78; *p* = 0.024). In the point comparison, only at 24 h do we see a difference in the multiple comparison adjustment (*p* = 0.008), with lower SBP in the cardiometabolic compared with the cardiac group (*p* = 0.007) but not the cardiorenal group (*p* = 0.067). Significant differences in mean arterial pressure (MAP), similar to SBP, were seen between groups (*F* = 3.52; *p* = 0.031), but only at 24 h did we see differences in the time-point analysis. Considering DBP, no differences were observed in the between- or within-subjects effects, but only at 24 h did the cardiac group show a higher DBP (*p* = 0.007). In addition, a difference in comparison with the cardiometabolic group was seen in the multiple comparisons (*p* = 0.01).

Perfusion pressure (MAP-RAP) showed significant group differences and higher *F* values than its derived components (*F* = 8.17; *p* < 0.001), with lower values in the cardiorenal and cardiometabolic groups in the multiple comparisons. However, all time points showed a difference after 6 h. In the pairwise comparison, we saw differences between the cardiac vs. the cardiometabolic group and at 24 h vs. the cardiorenal group.

RAP showed significant differences among the groups (*F* = 6.64; *p* = 0.001), with higher values in the cardiometabolic and cardiorenal groups for all time points. Significant differences were seen at all time points in the point data analysis. When corrected by multiple comparisons at 12 and 24 h, differences were seen between the cardiac and cardiorenal groups and at all time points for the cardiometabolic group. PCWP showed significant differences between groups (*F* = 4.71; *p* = 0.01), with the lowest values in the cardiac group. This was only significant in the time-point analysis at the 24 h mark.

Considering pulmonary artery systolic pressure (PASP), no difference was initially seen in the analyses of variance (ANOVAs) between subjects, but the values decreased over time, and lower values were seen in the cardiac group at the 24 h time-point analysis. However, the significance was not retained in the pairwise analysis. PA diastolic pressure (PADP) had differences in the group · ime interaction, with lower values in the cardiac group and a tendency to decrease compared with the increasing levels in the cardiorenal and cardiometabolic profiles. The point analysis showed differences only at 24 h, with the lowest values in the cardiac group, which, in the pairwise comparisons, only significantly differed from the cardiometabolic group. Considering the mean PA pressure (mPAP), no differences between subjects were seen in the ANOVA results. In the time-point analysis, differences were seen at 6 h, with the highest values in the cardiometabolic profile. At 24 h, the cardiac group had the lowest values, although no differences were seen in the pairwise analysis. Considering the pulmonary artery pressure index (PAPi), no differences were seen between the groups in ANOVA or time-point analyses.

Cardiac output, index, power, power index, and CPI_(RAP)_ showed similar behavior when using ANOVA, and differences arose in the between-group comparison (*p* < 0.05), with the lowest cardiac output and derived indexes in the cardiometabolic group and the highest levels in the cardiac group. In the time-point analysis, all times showed differences, with a pairwise comparison showing lower levels in cardiometabolic patients vs. cardiac patients; only at 6 h did the cardiorenal group show higher levels of cardiac output, index, power, and CPI_(RAP)_ than the cardiometabolic group. (Table 2, Figure 1 and Figure 2, and Appendix A)

Stroke volume (SV) and stroke volume index (SVi) data show differences in both SV and SVI (*F* = 6.91, *p* = 0.001 and *F* = 4.63, *p* = 0.01, respectively). In the time-point analysis, the cardiometabolic group had far worse hemodynamic parameters at baseline in the pairwise comparison with the cardiac and cardiorenal groups. However, differences were only seen in contrast with the cardiac group at 6 and 12 h.

Systemic vascular resistance (SVR) but not the SVR index (SVRi) had significant group differences in the ANOVA results (*F* = 3.1, *p* = 0.046). These differences were further enhanced in the time-point analysis, and only at 6 h did SVR and SVRi show some higher differences in the cardiometabolic group. However, neither pulmonary vascular resistance (PVR) nor PVR index (PVRi) did not achieve statistical differences among the groups. In the time-point analysis, only at 6 and 12 h did PVR have significant differences, with the lowest values in the cardiac group, and PVRi only achieved this at 6 h.

The left-ventricular stroke work index (LVSWi) showed differences (*F* = 7.18, *p* = 0.001), with the lowest values seen in the cardiometabolic group along the four-time points compared with the cardiac group. However, the right-ventricular stroke work index (RVSWi) did not show these differences. Only at the baseline were significant differences between the cardiometabolic and cardiac groups.

### 4.2. Hemodynamic Variables and Multi-Organ Failure, AKI, and Ventricular Arrhythmias 

Multi-organ failure: Regarding CPI_(RAP)_, lower values with statistical significance were seen in the MOF group. In the time-point analysis at baseline, 6, 12, and 24 h, lower perfusion pressure was seen in MOF. LVSWi had differences more marked in this group, with all these hemodynamic parameters significantly different at all time points (Appendix A).

AKI: CPI_(RAP)_ had significant differences overall lower in patients with AKI at all time points. SBP, MAP, and perfusion pressure were significant in the ANOVA, with lower values in patients with AKI. PCWP had substantial differences (*F* = 4.42, *p* = 0.036) with higher values in AKI, which was seen at 6 and 24 h in the time-point analysis. LVSWi had differences with lower values in AKI, but in the time-point analysis, this was seen from 6 to 24 h (Appendix A).

Ventricular tachycardia or fibrillation: Low CPI_(RAP)_, perfusion pressure, and LVSWi were seen in the ventricular tachycardia/fibrillation (VT/VF) in the first 24 h of active hemodynamic monitoring (Appendix A). (Table 3, for Detailed Description, See the Appendix A, Appendix A, Appendix A).

### 4.3. Survival Analysis

Group differences were significant in the Cox and log regression tests (Cox = 34.85, log-rank = 35.62; both *p* < 0.001). Considering the RMST, the cardiac group had more survival days than the cardiorenal and cardiometabolic groups at 6.04 (2.93–9.15; *p* < 0.001) and 10.63 (7.59–13.57; *p* < 0.001), respectively. When these last two groups were compared, the cardiorenal group had an advantage of 4.59 days (1.08–8.1; *p* = 0.01). The Cox regression showed an HR = 2.1 (1.4–3.14; *p* < 0.001) in the cardiorenal phenotype and 3.28 (2.19–4.92) in the cardiometabolic phenotype compared with the cardiac phenotype, with an AUC of 0.661 (0.606–0.714) for prediction power (Figure 3A,E).

In the case of complications, the presence of MOF showed differences (*p* < 0.001), with an HR = 1.94 (1.37–2.76; *p* < 0.001) and an RMST of 5.63 (2.95–8.31; *p* < 0.001) days less in the presence of MOF. Patients with AKI had an HR = 1.7 (1.16–2.49; *p* = 0.007). Furthermore, these patients had a survival disadvantage of 4.34 (1.44–7.24; *p* = 0.003) days, while patients with VT/VF had an HR = 2.15 (1.56–2.97; *p* < 0.001) and patients who did not present this complication had a survival advantage of 7.17 (4.35–9.99; *p* < 0.001) days. An increase of one point in the MODS score gave an HR = 1.1 (1.06–1.14; *p* < 0.001; Figure 3B–E).

### 4.4. Multivariate Analysis

When adjusted for SCAI and ∆congestion, the phenotyping retained its significance (*p* < 0.001); furthermore, SCAI and ∆congestion also appeared to have an independent value for mortality prediction (*p* = 0.011 and 0.028, respectively), with an AUC of 0.72 (0.67–0.77; Figure 3E).

When adjusted for the clinically relevant variables, CS phenotypes had a significant difference in the Cox regression (*p* = 0.002), with a cardiorenal HR = 1.74 (1.14–2.68; *p* = 0.011) and a cardiometabolic HR = 2.22 (1.4–3.51; *p* = 0.001). MOF had an adjusted HR = 1.56 (1.08–2.25; *p* = 0.017); for VT/VF, the HR = 1.93 (1.38–2.7; *p* < 0.001); in the case of AKI, a loss of statistical significance was seen, with an HR = 1.25 (0.84–1.85; *p* = 0.278). Finally, for MODS, significant differences were found, with an HR = 1.07 (1.03–1.12; *p* < 0.001) per point (Figure 3E).

## 5. Discussion

Herein, we describe the full invasive hemodynamic profiling of AMI-CS. To our knowledge, this is the first attempt to define a three-axis model of CS profiling (phenotype + SCAI + congestion). Previously, the proposed phenotypes by Zweck et al. [2] correctly classified specific higher mortality groups; we aimed to allocate the patient groups into specific phenotypes by utilizing alanine aminotransferase (ALT) and estimated glomerular filtration rate (eGFR) as straightforward indicators. Moreover, the independent mortality estimated by SCAI or change in congestion could help to allocate high-intensity therapies, such as MCS, or other resourceful interventions. We also demonstrated that complications, such as MOF, AKI, and VT/VF, increased mortality in AMI-CS.

CS phenotypes exhibit a distinct hemodynamic signature, with the cardiometabolic group demonstrating the worst hemodynamic parameters. Regarding congestion, previous studies have shown that patients in the cardiorenal and cardiometabolic groups have higher RAP and PCWP levels, indicating that they have trouble achieving decongestion compared with the cardiac group [2,3]. As expected, patients in the cardiac group had the highest cardiac output and power and their derived measures in our study, while patients in the cardiometabolic group had the lowest levels.

The best hemodynamic parameters to distinguish between the groups were cardiac power followed by CPI_(RAP)_. Baldetti et al. suggests that a cut-off of 0.28 W/m^2^ indicates an increased risk of mortality in a time-fixed manner [11]. We saw that the cardiometabolic group had more trouble achieving higher values than the other two groups. These findings, derived from perfusion pressure, suggest that increased congestion is observed in the cardiorenal and cardiometabolic groups, and they can identify more splanchnic damage (renal and liver) [11]. PADP showed a particular response based on the group type. Patients in the cardiac group had lower levels of PADP, while patients in the cardiorenal and cardiometabolic groups had increased levels as the first 24 h progressed. LVSWi also showed good discriminative power; previous studies suggested that it can better discriminate mortality risk than LVEF and improve mortality risk stratification [12]. This could aid in characterizing phenotypes, as a step up in LVSWi is observed in cardiorenal and cardiac groups in contrast to the cardiometabolic group.

Interestingly, the vasoactive analysis reveals intriguing association differences in usage among the different CS phenotypes. Vasopressin and dobutamine showed more pronounced disparities, with cardiometabolic patients often requiring higher percentages. Also, levosimendan showed a higher use in cardiorenal and cardiometabolic groups. These associations suggest that CS phenotypes may have varying hemodynamic needs, possibly linked to their underlying phenotype-related pathophysiology and the severity of organ involvement, which is usually more pronounced in the cardiometabolic group. Further research is needed to uncover the mechanisms behind these differences and their implications in CS management, as these associations probably underlie the higher MOF seen in the cardiorenal and cardiometabolic phenotypes.

MOF development could be discerned from the first 24 h hemodynamics. As suggested by previous studies, CPI_(RAP)_ had the best discriminative power, followed by perfusion pressure. An inadequate pressure–flow state, which is globally measured by CPI_(RAP)_ and impaired in MOF, compromises tissular metabolic demands, which leads to end-organ failure. In the CardShock study [13], variables such as confusion, elevated blood lactate, and eGFR were predictors of in-hospital mortality, as in the MODS system [10]. Thus, it is important to underscore the usefulness of CPI_(RAP)_ and also LVSWi as a hemodynamic goal and a discriminative power to identify patients who develop MOF [11,13,14]. Lower PAPi levels in MOF suggest that these patients had an overall worse RV function, which is also supported by the fact that these patients had more RV congestion, as seen by higher RAP levels [15]. Therefore, as proposed previously [4], an effective rapid decongestion is paramount to avoiding MOF.

The development of AKI has been associated with higher overall mortality. Unlike previous studies on the cardiorenal syndrome that have mainly focused on heart failure and that failed to find an association between cardiac index and AKI development [16,17], our study revealed that lower levels of cardiac power and output, as well as their derived measurements, had an impact on the development of AKI. In addition, CPI_(RAP)_ was found to have the best discriminative power, possibly because of the different hemodynamic responses in the acute setting of AMI-CS [11,14]. Finally, there were substantial differences in PCWP and RAP, with the latter observed not between groups but as an interaction with time. High LV congestion and ineffective RV/LV decongestion led to AKI development.

Few studies have investigated the relationship between hemodynamics and arrhythmogenesis in cases of electrical instability. Typically, the underlying mechanism is an ischemia-induced insult resulting in low pressures. The most useful parameter for predicting arrhythmogenesis, as with previous complications, is CPI_(RAP)_. While VT/FV has been extensively studied in advanced HF [18], there is a lack of research in patients with AMI-CS, highlighting the importance of predicting this complication. Our cohort shows low pressure–flow parameters among patients who develop these arrhythmias. Thus, achieving an adequate pressure–flow state is crucial, and the optimization by MCS or pharmacological treatment could potentially prevent arrhythmia development.

The cardiogenic shock profiling aids to provide a more granular classification of the classic types of shock (cardiac, hypovolemic, septic, etc.). In our study’s three-axis model for subsets of AMI-CS, profiling offers clinicians a tool to personalize treatments, optimizing resource allocation and ultimately improving patient outcomes. Also, axis phenotyping could help us design appropriate granular data to study patients that might benefit from MCS [3], especially VA-ECMO, since trials showed no reduction in 30-day outcomes in AMI-CS all-comers [19,20].

The current study’s limitations are its single-center retrospective data, the lack of a specific time for complications, and the lack of records of the particular vasoactive drug dose, the timing of MCS, and the response to PAC-derived hemodynamic data, which prevents the calculation of other scores, such as SOFA. In addition, as the cohort has inherent mortality or PAC withdrawal losses, the expectation–maximization algorithm’s intrinsic limitations impact the current data. Nevertheless, this method helps us to understand the hemodynamic trajectories and is more informative and statistically rigorous [21,22]. The strengths of the present study are the large cohort and the full record of the hemodynamic profiling in an academic center, which kept all specific primary PAC-derived data, which contrasts with the scarce complete PAC profiling for AMI-CS in a previous registry [5]. The longitudinal PAC measures described here, and the dynamic nature of AMI-CS could help improve our understanding of this high mortality entity, develop prevention strategies, and allocate resources more effectively.

## 6. Conclusions

Comprehensive phenotyping in AMI-CS can provide valuable patient-level prognostic information. The phenotyping of cardiogenic shock reveals varying mortality rates and complications. In addition, specific hemodynamic behaviors can signal potentially high-risk complications, such as MOF, AKI, and/or ventricular arrhythmias. Therefore, complete phenotyping in patients with AMI-CS is crucial for providing accurate prognosis and for the design of new trials.

## Figures and Tables

**Figure 1 jcm-12-05818-f001:**
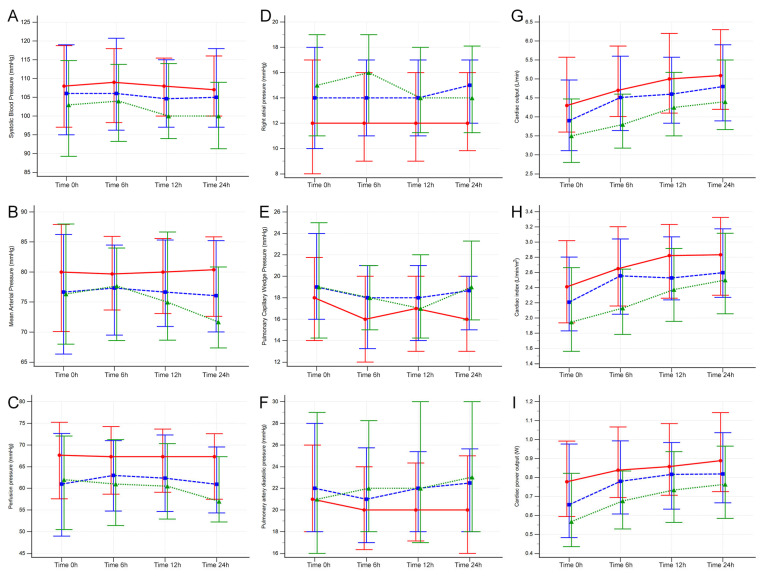
ANOVA repeated measures in CS-AMI phenotypes. Red—cardiac-only; blue—cardiorenal; and green—cardiometabolic; with median and interquartile ranges represented for systolic blood pressure (**A**), mean blood pressure (**B**), perfusion pressure (**C**), right atrial pressure (**D**), pulmonary capillary wedge pressure (**E**), pulmonary artery diastolic pressure (**F**), cardiac output (**G**), cardiac index (**H**), and cardiac power output (**I**). (Key in Figure 2).

**Figure 2 jcm-12-05818-f002:**
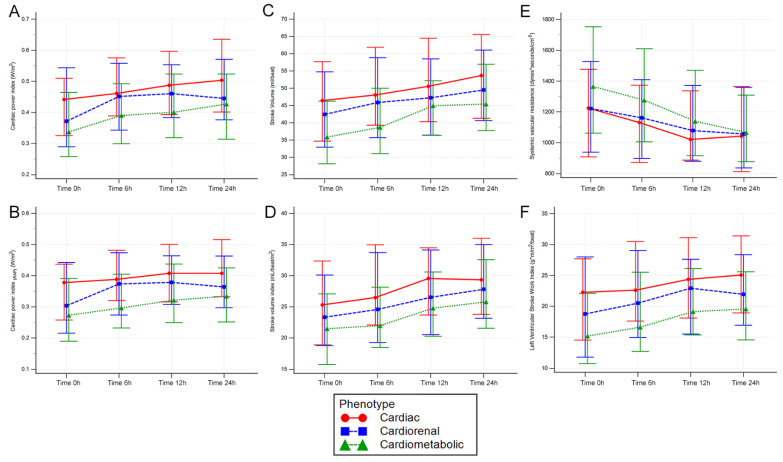
ANOVA repeated measures in CS-AMI phenotypes. Red—cardiac-only; blue—cardiorenal; and green—cardiometabolic; with median and interquartile ranges represented for cardiac power index (**A**), cardiac power index_(RAP)_ (**B**), stroke volume (**C**), stroke volume index (**D**), systemic vascular resistance (**E**), and left ventricular stroke work index (**F**). The key is shown at the bottom.

**Figure 3 jcm-12-05818-f003:**
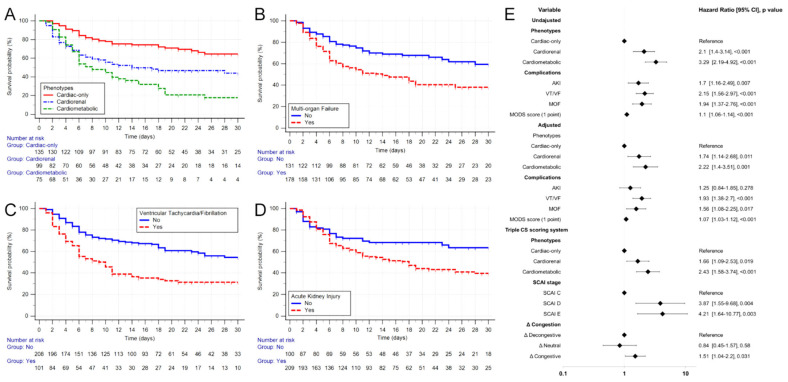
Kaplan–Meier curves according to phenotypes (**A**), multi-organ failure (**B**), ventricular tachycardia/fibrillation (**C**), and acute kidney injury (**D**). Forest plot for Cox regression (**E**).

**Table 1 jcm-12-05818-t001:** Characteristics of patients with acute myocardial infarction cardiogenic shock phenotypes.

Variables		Cardiac-Only (*n* = 135)	Cardiorenal (*n* = 99)	Cardiometabolic (*n* = 75)	*p*-Value
Gender	Male	115 (85.2%)	78 (78.8)	61 (81.3)	0.439
	Female	20 (14.8%)	21 (21.2)	14 (18.7)	
Age (years)		57 (50–64)	64 (58–69)	62 (56–69)	<0.001
Weight (kg)		76 (68–85)	75 (65–85)	75 (68–80)	0.534
Height (m)		1.68 (1.61–1.7)	1.67 (1.6–1.71)	1.68 (1.6–1.7)	0.516
BMI (kg/m^2^)		27.49 (24.51–29.39)	26.86 (24.2–29.05)	26.64 (24.49–29.41)	0.683
BSA (m^2^)		1.8 (1.7–1.9)	1.8 (1.68–1.92)	1.77 (1.67–1.9)	0.129
Smoking history (%)		88 (65.2)	59 (59.6)	42 (56)	0.394
Hypertension (%)		57 (42.2)	59 (59.6)	44 (58.7)	0.012
COPD (%)		2 (1.5)	5 (5.1)	2 (2.7)	0.286
Previous HF episodes (%)		9 (6.7)	5 (5.1)	8 (10.7)	0.348
Previous CKD (%)		0	12 (12.1)	5 (6.7)	<0.001
Diabetes mellitus (%)		58 (43)	56 (56.6)	35 (46.7)	0.115
Previous MI (%)		25 (18.5)	13 (13.1)	11 (14.7)	0.51
Previous PCI (%)		8 (5.9)	8 (8.1)	6 (8)	0.772
Previous CABG (%)		2 (1.5)	1 (1)	3 (4)	0.494
Type of AMI (%)	NSTEMI	28 (20.7)	20 (20.2)	5 (6.7)	0.022
	STEMI	107 (79.3)	79 (79.8)	70 (93.3)	
Out-of-hospital cardiac arrest (%)		4 (3)	3 (3)	3 (4)	0.92
Killip–Kimball (%)	I	31 (23)	10 (10.1)	5 (6.7)	<0.001
	II	55 (40.7)	21 (21.2)	13 (17.3)	
	III	33 (24.4)	30 (30.3)	13 (17.3)	
	IV	16 (11.9)	38 (38.4)	44 (58.7)	
LVEF (%)		35 (30–43)	32 (26–40)	30 (20–45)	0.073
Heart rate (bpm)		96 (85–110)	97 (84–110)	101 (81–117)	0.552
SBP (mmHg)		108 (97–119)	106 (95–119)	103 (89–115)	0.143
DBP (mmHg)		65 (56–74)	60 (52–72)	65 (54–75)	0.25
Hemoglobin (g/dL)		14.5 (12.6–16.3)	14 (12.1–16)	14.2 (12.8–16.3)	0.756
WBC (cell/mm^3^)		12 (9.4–15.4)	13.6 (10.5–17.5)	14.7 (11.8–18.5)	<0.001
Neutrophils (%)		79.35 (71.5–85.1)	82.75 (78–86.6)	82 (76.6–86)	0.024
Lymphocytes (%)		13.25 (8.6–17.1)	10.5 (7.2–15)	11 (7–16.5)	0.024
Platelets (cell/mm^3^)		224 (179–287)	241 (194–285)	195 (144–274)	0.005
PT (sec)		13.5 (12.1–15)	14 (12.4–16.1)	13.9 (12–18)	0.154
PTT (sec)		33.6 (29.8–39)	35.25 (30.1–45)	34.2 (30.1–43)	0.37
Glucose (mg/dL)		153 (128–248)	213 (130–324)	217 (146–308)	0.007
BUN (mg/dL)		18 (13–24)	30 (23–40.8)	35.21 (24–50)	<0.001
Creatinine (mg/dL)		1 (0.9–1.1)	1.8 (1.5–2.5)	2 (1.5–2.61)	<0.001
eGFR by CKD-EPI (mL/min/m^2^)		81.78 (69.59–95.14)	39.18 (26.59–49.5)	32.1 (24.38–47.84)	<0.001
Chloride (mEq/L)		104 (101–107)	104 (100–108)	102 (98–105)	0.04
Sodium (mEq/L)		136 (133–138)	136 (132–140)	136 (133–139.1)	0.859
Potassium (mEq/L)		4.1 (3.8–4.4)	4.6 (4.2–5.2)	4.4 (4.09–5.2)	<0.001
Magnesium (mg/dL)		2.28 (2–2.5)	2.3 (2.04–2.5)	2.1 (2–2.33)	0.131
Albumin (g/dL)		3.4 (2.99–3.88)	3.27 (2.9–3.7)	3.3 (2.9–3.6)	0.159
AST (U/L)		76.5 (41–249)	83.35 (40–229)	700 (325–992)	<0.001
ALT (U/L)		61.5 (34.8–103.5)	51.3 (36.2–84.5)	450 (204.4–1257)	<0.001
LDH (U/L)		635 (330–1248)	545 (324–1244)	1290 (715–2250)	<0.001
Hs-C-reactive protein (mg/L)		63.25 (21.2–130.5)	56.8 (20.2–153)	102.3 (35.4–150.8)	0.353
Maximum creatinine (mg/dL)		1.1 (0.99–1.48)	2.3 (1.7–3.19)	2.94 (1.93–3.74)	<0.001
Maximum AST (U/L)		129 (56.9–406)	128 (54–424)	832 (465–1951)	<0.001
Maximum ALT (U/L)		68.75 (37.95–111.85)	54.85 (36.4–84.7)	468.3 (210–1257)	<0.001
Bilirubin (mg/dL)		0.96 (0.66–1.61)	0.99 (0.54–1.58)	1.46 (0.96–2.37)	0.006
Minimum PaO_2_/FiO_2_ ratio		247.62 (146–304.76)	200 (120–300)	152.8 (100–257)	0.004
Lactate		2.1 (1.45–3.3)	2.95 (1.7–6)	4.35 (2.8–6)	<0.001
Base excess		−5.19 (−8.21; −3.12)	−8.78 (−13.23; −5.51)	−10.66 (−14.87; −7.59)	<0.001
pH		7.37 (7.32–7.42)	7.29 (7.23–7.37)	7.28 (7.22–7.35)	<0.001
Minimum 24 h MAP		72 (65.33–78.33)	67.67 (61.57–74)	68 (60–72)	0.003
Minimum 24 h SBP		98 (90–104)	96 (86–101)	91 (81–102)	0.01
Time of delay symptom to ER (hr:min)		12:16 (5:05–60:39)	16:21 (7:00–58:37)	28:42 (11:50–47:00)	0.253
Primary reperfusion (<12 h, %)	PI	27 (20.1)	17 (17.2)	5 (6.7)	0.005
	PCI	47 (34.8)	27 (27.3)	16 (21.3)	
	NR	61 (45.5)	55 (55.6)	54 (72)	
Angiography (%)		126 (93.3)	78 (78.8)	67 (89.3)	0.003
Total PCI (%)		103 (76.3)	59 (59.6)	53 (70.7)	0.023
Mechanical ventilation (%)		82 (60.7)	72 (72.7)	60 (80)	0.01
Hemodialysis (%)		6 (4.4)	17 (17.2)	21 (28)	<0.001
Norepinephrine (%)		100 (74.1)	82 (82.8)	66 (88)	0.039
Vasopressin (%)		62 (45.9)	56 (56.6)	55 (73.3)	0.001
Dobutamine (%)		103 (76.3)	83 (83.8)	73 (97.3)	<0.001
Levosimendan (%)		35 (25.9)	27 (27.3)	36 (48)	0.002
Number of vasoactive drugs (%)	None	17 (12.6)	4 (4)	1 (1.3)	<0.001
	1	17 (12.6)	15 (15.2)	3 (4)	
	2	39 (28.9)	24 (24.2)	13 (17.3)	
	3	43 (31.9)	39 (39.4)	31 (41.3)	
	4	19 (14.1)	17 (17.2)	27 (36)	
Mechanical support (%)		72 (53.3)	57 (57.6)	48 (64)	0.325
Acute kidney injury (%)		58 (43)	82 (82.8)	69 (92)	<0.001
AKIN stage (%)	None	77 (57)	17 (17.2)	6 (8)	<0.001
	1	32 (23.7)	33 (33.3)	24 (32)	
	2	13 (9.6)	18 (18.2)	12 (16)	
	3	13 (9.6)	31 (31.3)	33 (44)	
VT/VF (%)		44 (32.6)	27 (27.3)	30 (40)	0.208
Multi-organ failure (%)		56 (41.5)	62 (62.6)	60 (80)	<0.001
Number of organ failures (%)	0–1	79 (58.5)	37 (37.4)	15 (20)	<0.001
	2–3	47 (34.8)	44 (44.4)	35 (46.7)	
	4–5	9 (6.7)	18 (18.2)	25 (33.3)	
MODS score	0–4	66 (48.9)	31 (31.3)	10 (13.3)	<0.001
	5–10	49 (36.3)	36 (36.4)	26 (34.7)	
	≥11	20 (14.8)	32 (32.3)	39 (52)	
SCAI score	C	31 (23)	10 (10.1)	0	<0.001
	D	76 (56.3)	45 (45.5)	31 (41.3)	
	E	28 (20.7)	44 (44.4)	44 (58.7)	
Mortality (%)		45 (33.3)	52 (52.5)	53 (70.7)	<0.001

AKI: Acute Kidney Injury; ALT: Alanine Transaminase; AMI: Acute Myocardial Infarction; AST: Aspartate Transaminase; BMI: Body Mass Index; BSA: Body Surface Area; BUN: Blood Urea Nitrogen; CABG: Coronary Artery Bypass Grafting; CKD: Chronic Kidney Disease; COPD: Chronic Obstructive Pulmonary Disease; DBP: Diastolic Blood Pressure; eGFR: Estimated Glomerular Filtration Rate; HF: Heart Failure; Hs-CRP: High-sensitivity C-reactive Protein; LVEF: Left-Ventricular Ejection Fraction; LDH: Lactate Dehydrogenase; MAP: Mean Arterial Pressure; MI: Myocardial Infarction; MODS: Multiple Organ Dysfunction Syndrome; NSTEMI: Non-ST-Elevation Myocardial Infarction; NR: Non-primary Reperfusion; PCI: Percutaneous Coronary Intervention; PI: Pharmacoinvasive Strategy; PaO_2_/FiO_2_: Ratio of Arterial Oxygen Partial Pressure to Fraction of Inspired Oxygen; PT: Prothrombin Time; PTT: Partial Thromboplastin Time; SBP: Systolic Blood Pressure; SCAI: Society for Cardiovascular Angiography & Interventions; sec: seconds; STEMI: ST-Elevation Myocardial Infarction; VT/VF: Ventricular Tachycardia/Fibrillation; WBC: White Blood Cell Count.

**Table 2 jcm-12-05818-t002:** Hemodynamic parameters in AMI-CS phenotypes.

Hemodynamic Parameter	Cardiac-Only (*n* = 135)	Cardiorenal (*n* = 99)	Cardiometabolic (*n* = 75)	*p*-Value
Time 0 h				
Heart rate (bpm)	96 (85–110)	97 (84–110)	101 (81–117)	0.552
SBP (mmHg)	108 (97–119)	106 (95–119)	103 (89–115)	0.143
DBP (mmHg)	65 (56–74)	60 (52–72)	65 (54–75)	0.25
MAP (mmHg)	80 (70–88)	76.67 (66.3–86.33)	76.33 (68–88)	0.247
RAP (mmHg)	12 (8–17)	14 (10–18)	15 (11–19)	0.02
PCWP (mmHg)	18 (14–22)	19 (16–24)	19 (14–25)	0.068
PASP (mmHg)	36 (29–43)	37 (30–46)	37 (28–45)	0.71
PADP (mmHg)	21 (18–26)	22 (18–28)	21 (16–29)	0.611
mPAP (mmHg)	26 (21–31)	29 (22–34)	27 (21–35)	0.199
Cardiac output (L/min)	4.3 (3.6–5.6)	3.9 (3.1–5)	3.5 (2.8–4.5)	<0.001
Cardiac index (L/min/m^2^)	2.41 (1.93–3.02)	2.21 (1.82–2.81)	1.95 (1.56–2.67)	0.002
Cardiac power (W)	0.78 (0.59–0.99)	0.66 (0.48–0.98)	0.57 (0.43–0.83)	<0.001
Cardiac power index (W/m^2^)	0.44 (0.32–0.51)	0.37 (0.29–0.54)	0.34 (0.25–0.47)	0.002
CPI_(RAP)_ (W)	0.38 (0.26–0.44)	0.3 (0.21–0.45)	0.27 (0.19–0.39)	0.001
Perfusion pressure (mmHg)	67.67 (57.33–75.33)	61 (49–72.67)	62 (50.33–72.33)	0.035
PAPi	1.13 (0.67–1.77)	0.93 (0.61–1.5)	0.89 (0.65–1.48)	0.114
Stroke volume (mL)	46.44 (34.67–57.74)	42.42 (32.95–55.95)	35.9 (28.1–46.43)	0.001
Stroke volume index (mL/m^2^)	25.32 (18.72–32.58)	23.33 (18.75–30.12)	21.49 (15.76–27.31)	0.003
SVR (dynes/sec/cm^−5^)	1224.69 (907.26–1476.92)	1221.18 (935.38–1529.95	1363.81 (1059.36–1752.38)	0.078
PVR (dynes/sec/cm^−5^)	160 (86.25–247.42)	163.64 (66.67–293.88)	185.71 (107.46–311.69)	0.243
SVRi (dynes/sec·m^2^/cm^−5^)	2163.09 (1699.2–2784.83)	2266.67 (1666.42–2741.78)	2372.83 (1792–2979.24)	0.187
PVRi (dynes/sec·m^2^/cm^−5^)	274.07 (166.39–421.05)	302.22 (115.2–495.16)	335.19 (185.14–529.87)	0.35
LVSWi (gm-m/m^2^/beat)	22.24 (14.49–27.77)	18.74 (11.74–28.05)	15.2 (10.65–22.09)	0.001
RVSWi (gm-m/m^2^/beat)	4.41 (2.29–7.53)	4.15 (2.04–7.32)	3.16 (1.6–5.61)	0.048
Time 6 h				
Heart rate (bpm)	98 (90–107)	98.25 (86.22–112)	100 (81–115)	0.971
SBP (mmHg)	109 (98–118)	106 (96–121)	104 (93–114)	0.117
DBP (mmHg)	66 (57–73)	62 (52.84–71)	62 (54–72)	0.175
MAP (mmHg)	79.67 (73.67–86)	77.33 (69.33–84.67)	77.67 (68.33–84)	0.134
RAP (mmHg)	12 (9–16)	14 (11–17)	16 (12–19)	<0.001
PCWP (mmHg)	16 (12–20)	18 (13–21)	18 (15–21)	0.089
PASP (mmHg)	34 (29–39)	35 (30–44)	36 (29–44)	0.132
PADP (mmHg)	20 (16–24)	21 (17–26)	22 (18–28)	0.074
mPAP (mmHg)	25 (20–28)	26 (22–31)	27 (22–34)	0.038
Cardiac output (L/min)	4.7 (4–5.9)	4.51 (3.62–5.6)	3.8 (3.18–4.6)	<0.001
Cardiac index (L/min/m^2^)	2.65 (2.16–3.21)	2.56 (2.05–3.05)	2.13 (1.78–2.66)	<0.001
Cardiac power (W)	0.84 (0.69–1.07)	0.78 (0.6–0.99)	0.68 (0.53–0.84)	<0.001
Cardiac power index (W/m^2^)	0.46 (0.39–0.58)	0.45 (0.34–0.56)	0.39 (0.3–0.49)	<0.001
CPI_(RAP)_ (W)	0.39 (0.32–0.48)	0.37 (0.27–0.48)	0.3 (0.23–0.41)	<0.001
Perfusion pressure (mmHg)	67.33 (58.67–74.33)	63 (54.67–71)	61 (51.33–71.33)	0.003
PAPi	1.13 (0.63–1.86)	1.15 (0.69–1.5)	0.92 (0.48–1.53)	0.09
Stroke volume (mL)	48.05 (39.26–61.96)	45.87 (35.63–58.97)	38.67 (30.94–50.62)	<0.001
Stroke volume index (mL/m^2^)	26.49 (22.07–34.97)	24.59 (19.08–33.82)	21.97 (18.46–28.22)	0.001
SVR (dynes/sec/cm^−5^)	1130.98 (869.57–1377.78)	1160.78 (897.35–1412.14)	1276.19 (1000–1630.48)	0.011
PVR (dynes/sec/cm^−5^)	138.27 (86.49–196.49)	150.94 (88.08–266.67)	172.97 (106.67–316.28)	0.017
SVRi (dynes/sec·m^2^/cm^−5^)	2008.33 (1613.54–2431.73)	2059.71 (1596.27–2470.83)	2284.8 (1739.15–2763.24)	0.041
PVRi (dynes/sec·m^2^/cm^−5^)	251.45 (160–328.21)	271.7 (162.16–427.33)	310.4 (182.86–560)	0.028
LVSWi (gm-m/m^2^/beat)	22.6 (17.54–30.54)	20.5 (14.93–29.11)	16.63 (12.51–25.56)	0.001
RVSWi (gm-m/m^2^/beat)	4.18 (2.49–6.15)	4.23 (2.15–6.81)	3.47 (1.37–5.89)	0.149
Time 12 h				
Heart rate (bpm)	96 (87–106)	99 (88–112)	98 (86–114)	0.543
SBP (mmHg)	108 (100–115)	105 (97–115)	100 (94–114)	0.155
DBP (mmHg)	66 (57–73)	62 (56–70)	62 (55–74)	0.147
MAP (mmHg)	80 (73–85.67)	76.67 (70.67–85.33)	74.96 (68.67–86.67)	0.118
RAP (mmHg)	12 (9–16)	14 (11–17)	14 (11–18)	0.006
PCWP (mmHg)	17 (13–20)	18 (14–21)	17 (14–22)	0.278
PASP (mmHg)	36 (28–41)	37 (31–48)	39 (29–48)	0.117
PADP (mmHg)	20 (17–24)	22 (18–26)	22 (17–30)	0.088
mPAP (mmHg)	25 (20–30)	27 (22–32)	27 (21–35)	0.092
Cardiac output (L/min)	5 (4.1–6.2)	4.6 (3.81–5.6)	4.25 (3.5–5.2)	0.002
Cardiac index (L/min/m^2^)	2.82 (2.25–3.24)	2.53 (2.24–3.07)	2.37 (1.95–2.93)	0.003
Cardiac power (W)	0.86 (0.71–1.09)	0.82 (0.63–0.99)	0.73 (0.56–0.94)	0.003
Cardiac power index (W/m^2^)	0.49 (0.39–0.6)	0.46 (0.38–0.56)	0.4 (0.32–0.53)	0.007
CPI_(RAP)_ (W)	0.41 (0.31–0.5)	0.38 (0.31–0.46)	0.32 (0.25–0.44)	0.002
Perfusion pressure (mmHg)	67.33 (59–73.67)	62.33 (54.67–72.33)	60.51 (52.67–70.33)	0.009
PAPi	1.16 (0.76–1.7)	1.17 (0.8–1.89)	1 (0.67–1.86)	0.517
Stroke volume (mL)	50.57 (40.2–64.58)	47.22 (36.39–58.54)	44.92 (36.07–52.17)	0.003
Stroke volume index (mL/m^2^)	29.55 (23.68–34.54)	26.54 (20.34–34.2)	24.8 (20.27–30.69)	0.008
SVR (dynes/sec/cm^−5^)	1021.14 (885.19–1341.14)	1078.79 (880–1377.39)	1139.39 (915.94–1474.51)	0.319
PVR (dynes/sec/cm^−5^)	136.36 (80–202.81)	150.94 (84.75–233.11)	183.61 (105.26–266.67)	0.05
SVRi (dynes/sec·m^2^/cm^−5^)	1860.55 (1629.4–2311.95)	1993.85 (1543.11–2426.67)	1928.91 (1665.23–2580.39)	0.521
PVRi (dynes/sec·m^2^/cm^−5^)	247.27 (152–379.23)	273.68 (147.69–417.28)	317.29 (184.32–466.67)	0.101
LVSWi (gm-m/m^2^/beat)	24.37 (17.93–31.18)	22.9 (15.24–27.7)	19.15 (15.28–26.19)	0.003
RVSWi (gm-m/m^2^/beat)	4.85 (3.2–6.75)	4.33 (2.32–7.6)	4.02 (2.04–7.14)	0.381
Time 24 h				
Heart rate (bpm)	96 (86–109)	97 (87–111)	99 (80–115)	0.867
SBP (mmHg)	107 (100–116)	105 (97–118)	100 (91–109)	0.008
DBP (mmHg)	65 (58–71)	61 (55–69)	59 (53–69)	0.007
MAP (mmHg)	80.37 (72.6–85.91)	76.07 (69.81–85.33)	71.67 (67.33–81.33)	0.002
RAP (mmHg)	12 (10–16)	15 (12–17)	14 (11–18)	0.006
PCWP (mmHg)	16 (13–20)	19 (15–20)	19 (16–24)	0.001
PASP (mmHg)	35 (29–42)	38 (31–50)	40 (30–50)	0.03
PADP (mmHg)	20 (16–25)	22 (18–26)	23 (18–30)	0.001
mPAP (mmHg)	25 (21–30)	28 (23–33)	27 (23–34)	0.025
Cardiac output (L/min)	5.09 (4.2–6.3)	4.8 (3.89–5.9)	4.4 (3.65–5.5)	0.021
Cardiac index (L/min/m^2^)	2.83 (2.29–3.33)	2.6 (2.27–3.18)	2.5 (2.05–3.14)	0.045
Cardiac power (W)	0.89 (0.72–1.15)	0.82 (0.67–1.04)	0.76 (0.58–0.97)	0.001
Cardiac power index (W/m^2^)	0.5 (0.4–0.64)	0.44 (0.38–0.57)	0.43 (0.31–0.53)	0.003
CPI_(RAP)_ (W)	0.41 (0.33–0.52)	0.36 (0.3–0.46)	0.33 (0.25–0.43)	0.001
Perfusion pressure (mmHg)	67.33 (57.34–72.67)	60.95 (54.33–69.67)	57 (52–67.67)	<0.001
PAPi	1.25 (0.85–1.79)	1.2 (0.73–1.81)	1.05 (0.64–1.62)	0.129
Stroke volume (mL)	53.68 (41.18–65.79)	49.47 (40.43–61.06)	45.45 (37.63–57.27)	0.047
Stroke volume index (mL/m^2^)	29.35 (23.65–36.11)	27.82 (23.09–35.28)	25.81 (21.37–32.58)	0.081
SVR (dynes/sec/cm^−5^)	1041.86 (813.05–1365.99)	1056.74 (834.57–1360)	1066.67 (871.79–1310.64)	0.825
PVR (dynes/sec/cm^−5^)	136.17 (84.21–209.84)	148.15 (92.41–216.75)	163.64 (103.63–215.38)	0.246
SVRi (dynes/sec·m^2^/cm^−5^)	1875.17 (1541.33–2380.26)	1946.06 (1491.67–2317.73)	1942.86 (1525.33–2261.35)	0.995
PVRi (dynes/sec·m^2^/cm^−5^)	253.33 (150.77–374)	267.52 (172.73–372.13)	278.98 (179.73–409.23)	0.42
LVSWi (gm-m/m^2^/beat)	25.04 (18.92–31.43)	21.94 (16.85–28.51)	19.54 (14.53–25.61)	0.002
RVSWi (gm-m/m^2^/beat)	4.95 (3.02–6.88)	4.8 (2.92–7.75)	4.34 (2.73–6.54)	0.63

Bpm: Beats per minute; SBP: Systolic blood pressure; DBP: Diastolic blood pressure; MAP: Mean arterial pressure; RAP: Right atrial pressure; PCWP: Pulmonary capillary wedge pressure; PASP: Pulmonary artery systolic pressure; PADP: Pulmonary artery diastolic pressure; mPAP: Mean pulmonary artery pressure; CPI: Cardiac power index; CPI_(RAP)_: Cardiac power index normalized by right atrial pressure; PAPi: Pulmonary artery pulsatility index; sec: seconds; SV: Stroke volume; SVi: Stroke volume index; SVR: Systemic vascular resistance; PVR: Pulmonary vascular resistance; SVRi: Systemic vascular resistance index; PVRi: Pulmonary vascular resistance index; LVSWi: Left-ventricular stroke work index; and RVSWi: Right-ventricular stroke work index.

**Table 3 jcm-12-05818-t003:** Hemodynamics parameters in AMI-CS complications.

Hemodynamic Parameter	No MOF (*n* = 131)	MOF (*n* = 178)	*p*-Value	No AKI (*n* = 100)	AKI (*n* = 209)	*p*-Value	No VT/VF (*n* = 208)	VT/VF (*n* = 101)	*p*-Value
Time 0 h									
Heart rate (bpm)	93 (84–109)	99 (84–113)	0.088	99 (86–113)	97 (82–110)	0.278	97 (82–110)	98 (88–113)	0.173
SBP (mmHg)	109 (99–124)	102 (90–115)	0.001	108 (96–121)	104 (93–116)	0.16	108 (95–119)	102 (90–115)	0.093
DBP (mmHg)	66 (57–73)	61 (52–74)	0.108	66 (56–74)	62 (53–74)	0.228	64 (55–75)	60 (50–71)	0.109
MAP (mmHg)	80 (71.33–88)	75.67 (65.67–86)	0.005	78.83 (70.33–88.17)	76.67 (68–86.67)	0.193	78.67 (70–88.17)	76.67 (66.67–84.33)	0.083
RAP (mmHg)	13 (9–17)	15 (10–18)	0.093	14 (10–18)	14 (10–18)	0.788	13 (10–18)	15 (10–18)	0.764
PCWP (mmHg)	17 (13–22)	19 (16–23)	0.035	18 (13–23)	19 (15–23)	0.082	18 (14–22)	19 (15–24)	0.071
PASP (mmHg)	36 (30–45)	37 (29–45)	0.477	36 (29–44)	37 (30–45)	0.611	36 (29–44)	38 (30–46)	0.532
PADP (mmHg)	21 (17–28)	22 (18–27)	0.966	20 (17–28)	22 (18–28)	0.593	21 (17–27)	23 (18–29)	0.433
mPAP (mmHg)	26.1 (21–34)	27 (22–33)	0.963	25.5 (21–32.5)	27 (22–34)	0.248	27 (21–33)	28 (22–35)	0.261
Cardiac output (L/min)	4.3 (3.54–5.6)	3.83 (3–4.8)	0.001	4.35 (3.5–5.55)	3.9 (3.1–5)	0.024	4.2 (3.28–5.45)	3.81 (3–4.48)	0.012
Cardiac index (L/min/m^2^)	2.47 (1.94–3.11)	2.15 (1.69–2.72)	0.001	2.49 (1.93–3.11)	2.21 (1.78–2.76)	0.02	2.4 (1.86–3.11)	2.14 (1.73–2.52)	0.004
Cardiac power (W)	0.79 (0.62–0.99)	0.61 (0.47–0.9)	<0.001	0.79 (0.54–1)	0.66 (0.48–0.9)	0.025	0.73 (0.52–1)	0.64 (0.48–0.84)	0.003
Cardiac power index (W/m^2^)	0.45 (0.34–0.53)	0.35 (0.27–0.49)	<0.001	0.45 (0.31–0.53)	0.38 (0.28–0.49)	0.021	0.43 (0.3–0.53)	0.36 (0.27–0.45)	0.001
CPI_(RAP)_ (W)	0.38 (0.27–0.45)	0.28 (0.21–0.41)	<0.001	0.38 (0.23–0.44)	0.3 (0.22–0.41)	0.049	0.36 (0.23–0.46)	0.27 (0.21–0.38)	0.001
Perfusion pressure (mmHg)	68 (58–75.67)	62 (50–72.33)	0.002	65 (53.83–74)	64.33 (52.67–73.67)	0.37	65.33 (53.33–75.33)	64.33 (51–71.33)	0.17
PAPi	1.08 (0.67–1.75)	0.94 (0.63–1.58)	0.109	1 (0.64–1.74)	1 (0.65–1.6)	0.925	1 (0.65–1.7)	1 (0.67–1.5)	0.854
Stroke volume (mL)	46.43 (36.82–58.1)	38 (29.19–51.22)	0.001	44.81 (33.14–56.41)	41.12 (31.06–54.62)	0.255	45.08 (32.95–57.51)	37.47 (29.39–47.63)	0.002
Stroke volume index (mL/m^2^)	25.36 (20.57–32.77)	22.44 (16.77–28.79)	<0.001	24.41 (18.18–30.67)	23.21 (17.26–30.13)	0.293	24.71 (18.74–31.7)	21.03 (16.33–27.49)	0.001
SVR (dynes/sec/cm^−5^)	1224.24 (904.76–1575.38)	1276.61 (1026.67–1535.63)	0.413	1205.86 (900.25–1476.92)	1300 (1019.05–1574.6)	0.08	1249.95 (933.46–1519.69)	1236.3 (1052.99–1608.89)	0.297
PVR (dynes/sec/cm^−5^)	163.64 (88.89–266.67)	167.45 (102.86–260.16)	0.816	166.67 (78.13–254.83)	166.67 (102.86–266.67)	0.757	162.99 (88.96–253.66)	175 (96–293.88)	0.213
SVRi (dynes/sec·m^2^/cm^−5^)	2215.38 (1662.22–2818.33)	2260.34 (1813.33–2845.33)	0.404	2106.44 (1646.41–2602.21)	2278.6 (1792–2871.11)	0.058	2224.9 (1673.51–2801.78)	2270.12 (1851.71–2948.41)	0.207
PVRi (dynes/sec·m^2^/cm^−5^)	294.85 (160–493.71)	304.79 (180.4–453.33)	0.903	304.79 (159.22–441.35)	298.54 (178.18–493.71)	0.793	300.38 (175.11–454.56)	310.48 (170.67–528.81)	0.214
LVSWi (gm-m/m^2^/beat)	22.91 (15.47–28.38)	15.84 (11.45–24.53)	<0.001	21.11 (13.38–28.28)	17.8 (12.6–26.18)	0.117	21.47 (13.86–28.22)	14.61 (11.21–23.85)	<0.001
RVSWi (gm-m/m^2^/beat)	4.52 (2.01–7.72)	3.69 (2.15–6.01)	0.026	3.92 (1.72–7.23)	4.11 (2.29–6.91)	0.762	4.28 (2.03–7.23)	3.83 (2.29–6.88)	0.742
Time 6 h									
Heart rate (bpm)	98 (88–110)	98.85 (88–111)	0.817	100 (90.9–111.5)	98 (86–110)	0.099	98 (88–110)	99 (88–110)	0.581
SBP (mmHg)	109 (100–120.73)	105 (95–116)	0.012	108.96 (99.5–120.37)	105 (95–117)	0.075	107.49 (99–119.91)	104 (95–117)	0.079
DBP (mmHg)	66 (57–73)	61 (54–71)	0.058	66 (56.77–71)	62 (55–71)	0.342	65.5 (56.77–73.5)	60 (53–70)	0.018
MAP (mmHg)	80 (73.67–86.67)	77.17 (69.33–84)	0.007	80.5 (71.81–86.46)	77.67 (70–84.33)	0.081	79.33 (71.15–85.83)	76 (69.33–84)	0.014
RAP (mmHg)	13 (10–17)	14 (10–18)	0.167	13 (10–16)	14 (10–18)	0.026	14 (10–18)	14 (10–17)	0.731
PCWP (mmHg)	17 (12–22)	18 (14–20)	0.135	16 (12–20)	18 (14–21)	0.007	17 (13–21)	17 (15–21)	0.618
PASP (mmHg)	34 (28–42)	35 (30–42)	0.562	33 (28–40)	36 (30–43)	0.012	35 (29–42)	35 (30–42)	0.707
PADP (mmHg)	20 (16–24)	21 (17–26)	0.08	20 (16–24)	21 (17–26)	0.033	20 (17–25)	21 (17–25)	0.233
mPAP (mmHg)	25 (21–30)	26 (22–30)	0.386	24 (20–28)	26 (22–31)	0.004	25 (21–30)	36 (22–31)	0.117
Cardiac output (L/min)	4.7 (3.91–5.77)	4.14 (3.6–5.2)	0.002	4.8 (3.89–5.9)	4.24 (3.6–5.3)	0.005	4.56 (3.7–5.7)	4.1 (3.6–5.1)	0.063
Cardiac index (L/min/m^2^)	2.71 (2.28–3.15)	2.34 (1.96–2.91)	0.001	2.71 (2.2–3.26)	2.4 (2.03–2.94)	0.004	2.58 (2.07–3.17)	2.37 (1.96–2.76)	0.023
Cardiac power (W)	0.84 (0.68–1.06)	0.74 (0.56–0.93)	0.001	0.85 (0.69–1.1}	0.76 (0.59–0.93)	0.003	0.82 (0.64–1.05)	0.73 (0.59–0.9)	0.01
Cardiac power index (W/m^2^)	0.48 (0.39–0.58)	0.4 (0.32–0.53)	<0.001	0.5 (0.37–0.59)	0.41 (0.33–53)	0.003	0.46 (0.36–0.58)	0.4 (0.32–0.5)	0.003
CPI_(RAP)_ (W)	0.4 (0.32–0.49)	0.33 (0.26–0.44)	<0.001	0.41 (0.31–0.52)	0.35 (0.27–0.44)	0.001	0.39 (0.28–0.49)	0.33 (0.26–0.41)	0.001
Perfusion pressure (mmHg)	67 (58.33–75.67)	61.88 (55.33–71)	0.002	67.65 (58.17–75.33)	62 (55.33–72)	0.009	66.5 (57–74)	61.33 (54–69.67)	0.007
PAPi	1.18 (0.6–1.91)	1.07 (0.63–1.41)	0.114	1.08 (0.62–1.65)	1.12 (0.63–1.6)	0.846	1.11 (0.64–1.72)	1.1 (0.58–1.54)	0.312
Stroke volume (mL)	48.75 (38.91–61.26)	42.54 (34.56–57.14)	0.008	47.65 (38.95–61.88)	43.75 (35.66–57.66)	0.085	47.1 (35.97–61.76)	43.41 (36.13–53.54)	0.062
Stroke volume index (mL/m^2^)	27.24 (21.79–34.29)	23.28 (18.83–32.19)	0.006	26.55 (21.06–34.73)	23.53 (19.57–32.79)	0.113	25.73 (20.3–34.07)	22.85 (19.76–30.07)	0.021
SVR (dynes/secs/cm^−5^)	1125.93 (869.57–1377.78)	1200.23 (913.68–1500.95)	0.118	1131.34 (859.82–1394.2)	1190.7 (945.74–1468.64)	0.145	1168.51 (901.64–1458.96)	1171.93 (916.36–1422.22)	0.902
PVR (dynes/sec/cm^−5^)	145.45 (88.08–222.22)	156.67 (97.56–239.81)	0.296	131.8 (80.27–204.76)	158.84 (95.52–240)	0.096	139.36 (82.69–209.11)	172.55 (102.56–266.67)	0.017
SVRi (dynes/sec·m^2^/cm^−5^)	2000 (1627.69–2423.19)	2122.67 (1761.82–2581.9)	0.132	1980.2 (1567.28–2432.84)	2117.87 (1761.82–2544.57)	0.111	2057.57 (1625.84–2512.94)	2117.87 (1741.22–2470.83)	0.653
PVRi (dynes/sec·m^2^/cm^−5^)	256.6 (160–406.76)	273.12 (171.94–443.48)	0.31	239.75 (157.42–360.72)	274.54 (174.55–438.3)	0.114	252.11 (151.3–362.51)	304 (193.85–469.24)	0.012
LVSWi (gm-m/m^2^/beat)	24.26 (17.18–30.54)	18.97 (14.42–26.06)	<0.001	23.85 (16.17–31.06)	20.38 (14.99–27.38)	0.017	22.35 (15.83–30.95)	19.34 (14.37–24.38)	0.004
RVSWi (gm-m/m^2^/beat)	4.66 (2.49–6.52)	3.74 (2.03–5.47)	0.054	3.9 (2.42–5.92)	4.24 (2.15–6.13)	0.841	4.13 (2.04–6.77)	4.01 (2.44–5.27)	0.867
Time 12 h									
Heart rate (bpm)	98 (86–110)	96 (87–112)	0.942	98 (87–111)	96 (88–111)	0.832	96 (88–111)	98 (87–113)	0.807
SBP (mmHg)	108 (100–119)	104 (95–114)	0.025	108 (100–116)	104 (96–115)	0.102	108 (99–120)	102 (94–110)	0.001
DBP (mmHg)	66 (58–74)	63 (56–71)	0.109	66 (55–72)	63 (56–72)	0.335	64 (57–74)	63 (55–69)	0.025
MAP (mmHg)	80 (73.33–87)	76.83 (70.33–85)	0.024	80 (72.67–86.83)	77 (70.67–85.33)	0.135	80 (72.67–87)	75 (69.33–82)	0.002
RAP (mmHg)	13 (9–16)	14 (11–18)	0.007	12.15 (10–16)	14 (10–17)	0.042	13 (10–17)	13 (10–16)	0.982
PCWP (mmHg)	17 (13–21)	17 (14–21)	0.411	17 (13–20)	17 (14–21)	0.139	17 (14–21)	18 (14–21)	0.437
PASP (mmHg)	36 (29–45)	37 (30–44)	0.811	35 (29–43)	37 (29–46)	0.177	37 (29–44)	36 (29–46)	0.863
PADP (mmHg)	21 (17–25)	22 (18–27)	0.716	21 (18–25)	22 (17–27)	0.38	22 (18–26)	21 (17–25)	0.848
mPAP (mmHg)	27 (21–31)	26 (21–32)	0.834	25 (20–30)	27 (22–32)	0.069	26 (21–32)	27 (20–32)	0.75
Cardiac output (L/min)	5 (4.17–6.08)	4.55 (3.6–5.5)	0.004	5.07 (4.13–6.2)	4.6 (3.8–5.4)	0.004	4.8 (3.9–5.9)	4.55 (3.7–5.38)	0.158
Cardiac index (L/min/m^2^)	2.84 (2.35–3.24)	2.47 (2.03–2.94)	0.001	2.9 (2.4–3.25)	2.52 (2.14–2.97)	0.001	2.72 (2.22–3.19)	2.52 (2.14–2.95)	0.081
Cardiac power (W)	0.89 (0.71–1.06)	0.78 (0.6–0.95)	0.002	0.86 (0.72–1.11)	0.79 (0.63–0.98)	0.007	0.85 (0.66–1.06)	0.78 (0.59–0.92)	0.005
Cardiac power index (W/m^2^)	0.49 (0.4–0.59)	0.43 (0.34–0.53)	0.001	0.49 (0.4–0.6)	0.44 (0.35–0.54)	0.004	0.49 (0.39–0.58)	0.41 (0.35–0.5)	0.002
CPI_(RAP)_ (W)	0.42 (0.32–0.5)	0.35 (0.27–0.44)	<0.001	0.42 (0.34–0.5)	0.36 (0.28–0.45)	0.002	0.41 (0.31–0.5)	0.34 (0.28–0.42)	0.001
Perfusion pressure (mmHg)	67.67 (60–75)	62.52 (54.33–71)	0.001	67 (59.41–75)	62.67 (55–72)	0.03	66.83 (57.83–74.33)	61.33 (54.33–69)	0.002
PAPi	1.23 (0.8–2)	1.06 (0.73–1.62)	0.058	1.17 (0.77–1.73)	1.13 (0.74–1.8)	0.653	1.14 (0.74–1.87)	1.14 (0.78–1.62)	0.81
Stroke volume (mL)	49.59 (41.01–62.2)	45.85 (36.07–58.33)	0.012	51.33 (38.64–65.1)	46.51 (38.02–57.97)	0.019	48.54 (38.76–61.59)	45.92 (37.78–56.96)	0.167
Stroke volume index (mL/m^2^)	29.22 (23.92–34.31)	25.39 (20.27–31.45)	0.005	29.63 (23.37–35.92)	25.96 (21.08–31.48)	0.011	27.21 (22.14–34.39)	25.45 (20.15–31.71)	0.116
SVR (dynes/sec/cm^−5^)	1075.71 (894.62–1320.07)	1080.89 (882.05–1422.22)	0.514	1020.91 (836.98–1310.45)	1106.67 (896.97–1410.97)	0.142	1074.89 (879.58–1388.89)	1079.07 (905.78–1320.07)	0.989
PVR (dynes/sec/cm^−5^)	145.45 (84.21–222.12)	162.36 (95.93–233.04)	0.368	138.59 (79.7–198.51)	160 (93.33–241.86)	0.088	145.45 (90.44–218.33)	163.64 (88.89–233.33)	0.54
SVRi (dynes/sec·m^2^/cm^−5^)	1860.55 (1629.4–2348.82)	1956.65 (1633.75–2442.22)	0.41	1851.3 (1569.23–2227.65)	1955.56 (1644–2464)	0.105	1882.14 (1641.49–2432.2)	1987.11 (1604.27–2365.71)	0.924
PVRi (dynes/sec·m^2^/cm^−5^)	257.14 (147–411.67)	294.64 (163.33–409.66)	0.353	255.32 (133.32–362.1)	288.26 (163.02–435.35)	0.098	259.71 (161.96–403.14)	292.17 (161.68–417.28)	0.557
LVSWi (gm-m/m^2^/beat)	23.7 (19.05–30.97)	20.66 (15.54–26.72)	0.002	25.27 (17.91–31.75)	21.63 (16.26–26.72)	0.006	23.47 (17.31–30.83)	20.57 (15.8–25.63)	0.002
RVSWi (gm-m/m^2^/beat)	5.08 (3.3–7.6)	4.03 (2.04–6.59)	0.003	4.84 (3.22–6.57)	4.34 (2.42–7.16)	0.435	4.81 (2.44–7.11)	4.33 (2.94–6.76)	0.699
Time 24 h									
Heart rate (bpm)	96 (85–110)	98 (87–114)	0.219	97 (86–111)	97 (85–110)	0.645	96 (87–110)	98 (85–111)	0.849
SBP (mmHg)	108 (98–121)	103 (92–113)	0.008	108 (100–118)	103 (93–114)	0.012	107 (98–116)	101 (91–114)	0.014
DBP (mmHg)	65 (58–71)	60 (55–69)	0.038	66 (58–72)	60 (55–69)	0.006	65 (57–70)	59 (54–67)	0.002
MAP (mmHg)	80.33 (72.33–87)	74.68 (68.97–83.33)	0.006	80.73 (72.67–87.67)	75.67 (68.77–83.67)	0.003	79.83 (71.33–86.33)	72.67 (67.09–81.33)	0.001
RAP (mmHg)	13 (10–16)	14 (11–17)	0.037	13 (10–16)	14 (11–17)	0.307	13 (10–17)	14 (10–17)	0.777
PCWP (mmHg)	16 (14–20)	18 (15–22)	0.006	16 (14–20)	18 (14–22)	0.04	17 (14–20)	18 (15–22)	0.151
PASP (mmHg)	36 (29–45)	38 (30–47)	0.464	35 (29–42)	38 (30–47)	0.218	36 (29–46)	38 (32–47)	0.119
PADP (mmHg)	20 (17–25)	22 (18–27)	0.078	20 (17–26)	22 (18–27)	0.243	20 (16–26)	23 (18–28)	0.054
mPAP (mmHg)	26 (22–31)	27 (22–33)	0.266	25 (22–30)	27 (23–33)	0.134	25 (22–32)	27 (24–32)	0.075
Cardiac output (L/min)	5.1 (4–6.28)	4.68 (3.8–5.65)	0.024	5.15 (4.2–6.3)	4.7 (3.84–5.86)	0.02	4.9 (4–6.15)	4.7 (3.9–5.6)	0.098
Cardiac index (L/min/m^2^)	2.89 (2.29–3.4)	2.54 (2.17–3.08)	0.01	2.84 (2.44–3.39)	2.55 (2.16–3.16)	0.012	2.77 (2.25–3.3)	2.54 (2.2–3.02)	0.073
Cardiac power (W)	0.89 (0.71–1.16)	0.8 (0.63–0.98)	0.002	0.88 (0.73–1.15)	0.8 (0.63–0.99)	0.002	0.86 (0.67–1.13)	0.77 (0.62–0.93)	0.011
Cardiac power index (W/m^2^)	0.5 (0.4–0.64)	0.44 (0.34–0.54)	0.002	0.49 (0.41–0.64)	0.44 (0.34–0.56)	0.002	0.48 (0.38–0.63)	0.44 (0.35–0.51)	0.006
CPI_(RAP)_ (W)	0.41 (0.34–0.53)	0.36 (0.28–0.46)	<0.001	0.41 (0.33–0.55)	0.37 (0.28–0.46)	0.002	0.4 (0.31–0.51)	0.36 (0.28–0.42)	0.002
Perfusion pressure (mmHg)	66.33 (57.34–74)	59.67 (54–69.67)	<0.001	66.71 (58.33–74.46)	60.67 (54–70.33)	0.002	65 (56.5–72.93)	58.33 (53.33–67.33)	0.001
PAPi	1.31 (0.86–1.82)	1.09 (0.71–1.67)	0.071	1.3 (0.85–1.72)	1.15 (0.75–1.76)	0.71	1.21 (0.77–1.76)	1.15 (0.75–1.76)	0.932
Stroke volume (mL)	53.75 (40.48–65.88)	46.87 (39.26–60.19)	0.021	52.14 (41.39–67.71)	48.6 (39.26–61.58)	0.112	51.35 (40.33–64.69)	47.73 (38.78–59.76)	0.191
Stroke volume index (mL/m^2^)	29.87 (23.65–36.11)	26.34 (22.49–33.22)	0.012	29.19 (23.57–36.57)	27 (22.65–33.54)	0.107	28.59 (22.8–35.73)	26.08 (23.4–32.52)	0.125
SVR (dynes/sec/cm^−5^)	1048.89 (826.8–1270.78)	1040.75 (835.9–1365.99)	0.86	1046.18 (821.08–1271.05)	1044.44 (838.1–1360)	0.813	1060.22 (821.08–1358.66)	1028.57 (851.74–1290.32)	0.793
PVR (dynes/sec/cm^−5^)	141.18 (91.43–220)	150.77 (92.41–212.68)	0.789	132.34 (78.84–216.47)	153.76 (101.59–213.33)	0.137	147.97 (87.67–212.98)	152.38 (105.66–213.9)	0.321
SVRi (dynes/sec·m^2^/cm^−5^)	1880 (1537.85–2288.9)	1948.51 (1522.58–2379.75)	0.681	1849.85 (1554.09–2316.28)	1946.06 (1491.67–2330.22)	0.779	1917.26 (1543.83–2359.52)	1885.4 (1525.33–2322.04)	0.852
PVRi (dynes/sec·m^2^/cm^−5^)	259.05 (159.3–394.28)	272 (170.31–366.31)	0.829	247.62 (149.93–391.8)	273.78 (177.82–374)	0.217	258.52 (155.43–385)	268 (197–371.59)	0.249
LVSWi (gm-m/m^2^/beat)	25.49 (19.17–32.47)	20.57 (15.92–27.83)	<0.001	25.2 (18.71–32.78)	21.35 (15.99–27.91)	0.005	24.29 (17.2–31.55)	20.24 (17.15–25.4)	0.004
RVSWi (gm-m/m^2^/beat)	5.14 (3.49–7.75)	4.6 (2.34–6.18)	0.02	4.99 (3.04–7.12)	4.8 (2.79–6.54)	0.433	4.83 (2.64–7.36)	4.91 (3.59–6.25)	0.948

AKI: Acute kidney injury, MOF: Multi-organ failure, VT/VF: Ventricular tachycardia/fibrillation, bpm: Beats per minute, SBP: Systolic blood pressure, DBP: Diastolic blood pressure, MAP: Mean arterial pressure, RAP: Right atrial pressure, PCWP: Pulmonary capillary wedge pressure, PASP: Pulmonary artery systolic pressure, PADP: Pulmonary artery diastolic pressure, mPAP: Mean pulmonary artery pressure, CPI: Cardiac power index, CPI_(RAP)_: Cardiac power index normalized by right atrial pressure, PAPi: Pulmonary artery pulsatility index, sec: seconds, SV: Stroke volume, SVi: Stroke volume index, SVR: Systemic vascular resistance, PVR: Pulmonary vascular resistance, SVRi: Systemic vascular resistance index, PVRi: Pulmonary vascular resistance index, LVSWi: Left ventricular stroke work index; RVSWi: Right ventricular stroke work index.

## Data Availability

The datasets used and/or analyzed during the current study are available from the corresponding author on reasonable request.

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
