# Peer review of "Invasive Phenoprofiling of Acute-Myocardial-Infarction-Related Cardiogenic Shock"

_jcm, 2023, doi:10.3390/jcm12185818_

Round 1

Reviewer 1 Report

Dear authors, I was reviewing with interest the manuscript entitled "Invasive Pheno-profiling of Acute Myocardial Infarction related Cardiogenic Shock". You deal with a highly interesting topic and a paradigm shift in the evaluation of CS. You investigated retrospectively more than 300 pts. with impressive detailed data about PAC hemodynamics. According to the timeliness of the topic you should consider the following: Two actual references have to be added (JACC Heart Fail. 2023 Jun 6;S2213-1779(23)00239-1; Lancet. 2023 Aug 25;S0140-6736(23)01607-0. doi: 10.1016/S0140-6736(23)01607-0). In the discussion section, the fact that different phenotypes of CS seem to need different types of catecholamines needs to be discussed. However, after minor revisions the manuscript is suitable.

Author Response

Dear authors, I was reviewing with interest the manuscript entitled "Invasive Pheno-profiling of Acute Myocardial Infarction related Cardiogenic Shock". You deal with a highly interesting topic and a paradigm shift in the evaluation of CS. You investigated retrospectively more than 300 pts. with impressive detailed data about PAC hemodynamics. According to the timeliness of the topic you should consider the following: Two actual references have to be added (JACC Heart Fail. 2023 Jun 6;S2213-1779(23)00239-1; Lancet. 2023 Aug 25;S0140-6736(23)01607-0. doi: 10.1016/S0140-6736(23)01607-0).

R= Thank you for your thoughtful and constructive feedback on our manuscript entitled "Invasive Pheno-profiling of Acute Myocardial Infarction related Cardiogenic Shock." We sincerely appreciate your interest in our work and the insightful suggestions you have provided.

R=We agree that the inclusion of recent references is crucial to ensuring the timeliness and relevance of our study. We have promptly addressed this suggestion by incorporating the following references into our manuscript:

As ref 3: Zweck, E.; Kanwar, M.; Li, S.; Sinha, S.S.; Garan, A.R.; Hernandez-Montfort, J.; Zhang, Y.; Li, B.; Baca, P.; Dieng, F.; et al. Clinical Course of Patients in Cardiogenic Shock Stratified by Phenotype. JACC Heart Fail 2023, doi:10.1016/J.JCHF.2023.05.007.

And as ref 19: Zeymer, U.; Freund, A.; Hochadel, M.; Ostadal, P.; Belohlavek, J.; Rokyta, R.; Massberg, S.; Brunner, S.; Lüsebrink, E.; Flather, M.; et al. Venoarterial Extracorporeal Membrane Oxygenation in Patients with Infarct-Related Cardiogenic Shock: An Individual Patient Data Meta-Analysis of Randomised Trials. Lancet 2023, doi:10.1016/S0140-6736(23)01607-0.

We have indeed included the following line in the discussion section: 'Also, axis phenotyping could help us design appropriate granular data to study patients that might benefit from MCS,[3] especially VA-ECMO, since trials showed no reduction in 30-day outcomes in AMI-CS all-comers.[19,20].'

This addition further reinforces the relevance of our axis phenotyping approach in guiding patient selection for MCS, particularly in the context of MCS, where previous trials have posed challenges in improving short-term outcomes in AMI CS patients. We believe that this reference enhances the clarity and significance of our discussion. Once again, we sincerely appreciate your valuable input, which has contributed to the refinement of our manuscript.

In the discussion section, the fact that different phenotypes of CS seem to need different types of catecholamines needs to be discussed. However, after minor revisions the manuscript is suitable for being published in JCM.

R= We sincerely appreciate the reviewer's insightful feedback and the opportunity to address this important aspect in our discussion. In response to the comment regarding the usage of different types of catecholamines among CS phenotypes, we have included a dedicated paragraph to discuss this intriguing phenomenon:

'An interesting vasoactive analysis reveals intriguing association differences in usage among different CS phenotypes. Vasopressin and dobutamine showed more pronounced disparities, with cardiometabolic patients often requiring higher percentages. Also, levosimendan showed a higher use in cardiorenal and cardiometabolic groups. This association suggests that CS phenotypes may have varying hemodynamic needs, possibly linked to their underlying phenotype-related pathophysiology and severity of organ involvement, usually more pronounced in the cardiometabolic group. Further research is needed to uncover the mechanisms behind these differences and their implications in CS management, as these associations probably underlie the higher MOF of cardiorenal and cardiometabolic phenotypes.'

This addition to our discussion underscores the significance studying in a prospective new trials this association of this retrospective study. We believe that this insight contributes valuable depth to our manuscript, aligning it with the reviewer's constructive suggestions and further enhancing its scientific merit. We thank the reviewer for their valuable input, which has greatly contributed to the refinement of our manuscript."

Reviewer 2 Report

I read with interest the manuscript by Ortega-Hernandez and coauthors, investigating the hemodynamic profiles of 3 different CS phenotypes. Authors concluded that cardiometabolic phenotype had the highest mortality and worst hemodynamic profile. Moreover authors identified 3 prognostic scores.

The article is overall well written; methodology generally sounds. Authors should be congratulated for the efforts of describe the full invasive hemodynamic of 3 different CS profiles.

In discussion, authors stated that the prognostic information could help to allocate high intensity resources based on CS type.

My suggestion however is to enphasize this concept, higlighting the impact of such a study in clinical practice.

Minor comments:

1. The first sentence of the abstract "Phenotyping has identified three CS-phenotypes..." is not clear and should be rephrased.

2. Please define abbreviation and acronyms when mentioned for the first time both in abstract and text (e.g. CS, SCAI, etc).

minor English editing required

Author Response

I read with interest the manuscript by Ortega-Hernandez and coauthors, investigating the hemodynamic profiles of 3 different CS phenotypes. Authors concluded that cardiometabolic phenotype had the highest mortality and worst hemodynamic profile. Moreover authors identified 3 prognostic scores.

The article is overall well written; methodology generally sounds. Authors should be congratulated for the efforts of describe the full invasive hemodynamic of 3 different CS profiles.

In discussion, authors stated that the prognostic information could help to allocate high intensity resources based on CS type.

  1. My suggestion however is to emphasize this concept, highlighting the impact of such a study in clinical practice.

R= We sincerely appreciate the reviewer's valuable input, and we believe that by highlighting the clinical relevance and transformative potential of our study we believe adding this paragraph to the discussion could aid in this objective: “The cardiogenic shock profiling aids in a more granular classification of the classic types of shock (cardiac, hypovolemic, septic, etc.). In our study's 3-axis model for the subset of the AMI-CS, profiling offers clinicians a tool to personalize treatments, opti-mizing resource allocation and ultimately improving patient outcomes.”

Minor comments:

  1. The first sentence of the abstract "Phenotyping has identified three CS-phenotypes..." is not clear and should be rephrased.

R= Thanks for valuable feedback, we have refined the introductory sentence of the abstract hoping to enhance clarity and precision. “Studies had identified three CS-phenotypes (cardiac-only, cardiorenal, and cardiometabolic).”

  1. Please define abbreviation and acronyms when mentioned for the first time both in abstract and text (e.g. CS, SCAI, etc).

R=We appreciate the reviewer's feedback and have taken the necessary steps to enhance the clarity of our manuscript. We have now ensured that all abbreviations and acronyms, such as CS and SCAI, are appropriately defined when first mentioned in both the abstract and the main text. This adjustment will significantly improve the readability and comprehension of our work for our readers. We thank the reviewer for their valuable input, which has contributed to the overall refinement of our manuscript

Comments on the Quality of English Language: minor English editing required

R= Thank you for pointing out the need for minor English editing. We will thoroughly review and edit the manuscript to ensure the quality of language and readability. Your feedback is greatly appreciated.

Reviewer 3 Report

The article “Invasive Pheno-profiling of Acute Myocardial Infarction Related Cardiogenic Shock” is devoted to a very relevant and promising topic. In my opinion, the article is written quite well and can be considered after removing a number of minor adjustments.

1. The meaning of the sentence “We found an increased hazard ratio (HR) of 2.1 and 3.3 for the cardiorenal and cardiometabolic compared to the cardiac-only, respectively (P<0.001)” in the abstract is not entirely clear.

2. The sentence “For the CS proposal was defined as a systolic pressure of £90 mmHg, the need for vasoactive or mechanical support (MCS), lactate ³2 mmol/L, and/or a cardiac index of £2.2 L/min/m2” should have a link.

3. It is recommended to track the use of all abbreviations and transcripts in the text, for example, cardiogenic shock in the line 138, or ALT, AST, MOF, MODS.

4. Pictures are difficult to read, you need to make clearer images

5. In the discussion, in my opinion, it is worth highlighting the fact that shocks could be divided into other phenotypes too, for example, hypovolemic, septic types..

Author Response

The article “Invasive Pheno-profiling of Acute Myocardial Infarction Related Cardiogenic Shock” is devoted to a very relevant and promising topic. In my opinion, the article is written quite well and can be considered after removing a number of minor adjustments.

  1. The meaning of the sentence “We found an increased hazard ratio (HR) of 2.1 and 3.3 for the cardiorenal and cardiometabolic compared to the cardiac-only, respectively (P<0.001)” in the abstract is not entirely clear.

R= thank you for the suggestion now the abstract line reads “We found a hazard ratio (HR) of 2.1 for the cardiorenal and 3.3 for cardiometabolic versus the cardiac-only phenotype (P<0.001).” hope this change improve the clarity of the statement and we believe this suggestion help with the readability of our manuscript

  1. The sentence “For the CS proposal was defined as a systolic pressure of £90 mmHg, the need for vasoactive or mechanical support (MCS), lactate ³2 mmol/L, and/or a cardiac index of £2.2 L/min/m2” should have a link.

R= We appreciate the reviewer's suggestion and have now included a reference [8] to support the sentence: (6) Jones, T.L.; Nakamura, K.; McCabe, J.M. Cardiogenic Shock: Evolving Definitions and Future Directions in Management. Open Heart 2019, 6, e000960, doi:10.1136/OPENHRT-2018-000960.

  1. It is recommended to track the use of all abbreviations and transcripts in the text, for example, cardiogenic shock in the line 138, or ALT, AST, MOF, MODS.

R=We appreciate the reviewer's feedback and have taken the necessary steps to enhance the clarity of our manuscript. We have now ensured that all abbreviations and acronyms, are appropriately defined when first mentioned in both the abstract and the main text. This adjustment will significantly improve the readability and comprehension of our work for our readers. We thank the reviewer for their valuable input, which has contributed to the overall refinement of our manuscript.

  1. Pictures are difficult to read, you need to make clearer images.

R=We appreciate the reviewer's feedback regarding the clarity of our images. We have taken the necessary steps to improve the quality and readability of the images in our manuscript (especially figure 3). The updated images are now presented with enhanced resolution and clarity to ensure that all relevant details are easily discernible. We believe that these improvements will significantly enhance the visual presentation of our findings, making them more accessible and informative to our readers.

  1. In the discussion, in my opinion, it is worth highlighting the fact that shocks could be divided into other phenotypes too, for example, hypovolemic, septic types.

R= We appreciate the reviewer's insightful suggestion to further emphasize the differentiation of shock types, including hypovolemic and septic shocks, in our discussion. In response, we have included the following statement to underscore this important aspect:

The cardiogenic shock profiling aids in a more granular classification of the classic types of shock (cardiac, hypovolemic, septic, etc.). In our study's 3-axis model for the subset of the AMI-CS, profiling offers clinicians a tool to personalize treatments, optimizing resource allocation and ultimately improving patient outcomes.

We thank the reviewer for this valuable input, which has contributed to the enrichment of our manuscript.